# Medication Intake as a Factor for Non-Initiation and Cessation of Breastfeeding: A Prospective Cohort Study in Greece during the COVID-19 Pandemic

**DOI:** 10.3390/children10030586

**Published:** 2023-03-18

**Authors:** Maria Tigka, Dimitra Metallinou, Christina Nanou, Zoi Iliodromiti, Alexandros Gryparis, Katerina Lykeridou

**Affiliations:** 1Department of Midwifery, University of West Attica, 12243 Athens, Greece; maria.tigka@gmail.com (M.T.);; 2Department of Obstetric Emergency, General and Maternity Hospital “Helena Venizelou”, 11521 Athens, Greece; 3Department of Neonatology, Aretaieio Hospital, Medical School, National and Kapodistrian University of Athens, 15772 Athens, Greece; 4Department of Speech and Language Therapy, University of Ioannina, 45500 Ioannina, Greece

**Keywords:** medication intake, factor, non-initiation, breastfeeding cessation, breastfeeding, COVID-19 pandemic

## Abstract

Pharmacological treatment may become a barrier for a mother’s breastfeeding goals. We aimed to investigate maternal medication intake as a factor for non-initiation and cessation of breastfeeding and the effect of professional counseling on maternal decision-making. Throughout 2020, 847 women were recruited from five healthcare institutions. Information was gathered prospectively with an organized questionnaire through interview during hospitalization and through telephone at 1, 3 and 6 months postpartum. Results revealed that from the 57 cases of breastfeeding cessation due to medication intake, only 10.5% received evidence-based counseling from a physician. Unfortunately, 68.4% (*n* = 39/57) of the participants ceased breastfeeding due to erroneous professional advice. The compatibility of medicines with breastfeeding was examined according to the Lactmed and Hale classification systems, which showed discrepancy in 8 out of 114 medicines used, while 17.5% and 13.2% of the medicines, respectively, were not classified. Educational level, employment at six months postpartum, mode of delivery, previous breastfeeding experience, medication intake for chronic diseases, physician’s recommendation and smoking before pregnancy were factors significantly correlated with breastfeeding discontinuation due to medication intake. The COVID-19 restrictions protected women from ceasing breastfeeding due to medication intake. Maternal and lactation consultancy should be strictly related to evidence-based approaches.

## 1. Introduction

Most women are aware that exclusive and prolonged breastfeeding is a key element for optimal maternal and neonatal health outcomes. However, in their efforts to establish and sustain breastfeeding, mothers may encounter many barriers with significant implications for their breastfeeding goals. Such barriers during lactation may be conditions requiring maternal pharmacological treatment, including chronic or acute diseases and emergency medical conditions that demand diagnostic imaging or even a surgical procedure. In such scenarios, the decision-making process for medication use during lactation should be on the basis of healthcare professionals’ (HPs) current knowledge and expertise in providing information hinged on scientific research about the breastfeeding benefits and the drug exposure risks via breast milk for the nursing child [1]. Since the majority of medicines have not been linked to adverse effects when used during lactation, and that even a temporary interruption of breastfeeding can create difficulties for the mother–child dyad, lactating women should be enlightened about reliable resources which refer to medicines during lactation and, finally, supported to share decision-making with their physician [2,3].

Sometimes women receive conflicting opinions about the compatibility of a medicine with breastfeeding. Therefore, they might make decisions based on inaccurate information that would be detrimental to their intentions to breastfeed [4]. Unfortunately, women who require medication during lactation often cease breastfeeding early as a result of inappropriate professional counseling or concern that they are harming their child [5,6]. Although many medicines are probably safe for use during lactation, evidence-based data are relatively limited to provide clinicians with decision support tools, primarily because lactating women and their children are considered vulnerable populations and drug development excludes these groups. Nevertheless, alternative methods, such as population-based pharmacokinetic modeling, are continuously being developed to enhance the safety of medication during lactation [7].

Efforts are also being made worldwide to seek more information on the safety of medicines during lactation. The Food and Drug Administration (FDA) published the Pregnancy and Lactation Labeling Rule (PLLR) in 2014, which requires pharmaceutical companies to provide detailed information on the compatibility of medicines with pregnancy and lactation [8]. Guidance on pharmacovigilance procedures for the use of medicines during pregnancy and lactation is available in the European Union. After a product has been approved for marketing, it is mandatory to collect reports during the post-authorization phase on adverse reactions that may occur in breastfeeding infants. Proper collection and evaluation of data will provide patients and HPs with the necessary information to make informed decisions about using medications during lactation [9].

The promotion of breastfeeding depends on the implementation of national policies and recommendations at all levels of the health and social system to establish breastfeeding as the normal nutrition for infants and children [10]. Most European countries have established centralized phone-based qualified medical information on the potential toxicity of medications, where healthcare personnel and mothers can obtain information around the clock. In Greece, there is a national program named “Alkyoni” implemented by the Directorate of Social and Developmental Pediatrics of the Institute of Child Health, which aims at raising public awareness, supporting breastfeeding mothers and informing HPs. “Alkyoni” operates a nationwide telephone line since 2013 which is staffed by experienced midwives and pediatricians [11]. Alternative breastfeeding support helplines for the safety of medicines during lactation include the breastfeeding department of the “breastfeeding-friendly” public hospital “Helena Venizelou” [12].

All patient populations, including nursing mothers, have been affected by the COVID-19 pandemic. The WHO estimated that COVID-19 can be characterized as a pandemic on the 11th of March 2020 [13] and the Greek authorities implemented quarantine and social isolation measures a few days later, on the 16th of March. Research on human milk samples was immediately initiated to clarify whether SARS-CoV-2 is transmitted through breastfeeding; however, scientific evidence that this is highly unlikely was published on the 19th of August 2020. Nearly half a year of uncertainty had increased fear-based confusion, misinformation and ultimately enhanced the risk of breastfeeding discontinuation [14]. At the same time, limited access to health care services due to the COVID-19 restrictions might have affected the well-being of pregnant and lactating women [15] and might have further influenced medication use.

Recently, Ceulemans et al. in a multinational study, reported for the first time the prevalence and type of medications used by pregnant and breastfeeding women during the COVID-19 pandemic [16]. Drug utilization studies, particularly during social and health crises, provide critical insight to HPs and thus the aim of the present study was to determine (a) the types and safety of medications used by puerperal women during the COVID-19 pandemic and (b) the effect of professional counseling on maternal decision to initiate or discontinue breastfeeding when the need of medication intake arose. In the present study, ‘medication intake’ refers to prescribed and over-the-counter (OTC) medication.

## 2. Materials and Methods

### 2.1. Study Design and Ethical Considerations

The present study has a prospective longitudinal design. Data were gathered as a part of a wider cohort study in the midst of the COVID-19 pandemic [17]. Women were recruited between January and December 2020 from five tertiary maternity hospitals located in Attica, the capital of Greece. The three hospitals belong to the public sector, serve as referral hospitals and have a full complement of services, including neonatology, obstetrics and intensive care units. The other two hospitals belong to the private sector and offer the same services as the public ones. According to the Declaration of the World Medical Association of Helsinki, the main principles to justify this research were met. Ethics approval was granted by the Committees of all hospitals.

General and Maternity Hospital “HELENA VENIZELOU” (24285/29 October 2019)“ATTIKON” General University Hospital (570/1 October 2019)“ALEXANDRA” General Hospital (511/20 July 2020)“IASO” General Maternity and Gynecology Clinic (30 May 2019)“LETO” General, Maternity and Gynecology Clinic (174a/5 June 2019)

### 2.2. Sample and Setting

The sample and setting of the research study have been extensively described in a previous published article [17] and are summarized here as follows:

The study enrolled mothers who had given birth and were hospitalized in the postnatal ward. To increase the study’s accuracy, stratified random sampling was employed. Due to major pandemic constraints, recruitment should have been completed within two months in each maternity hospital. Thus, these specific time restrictions had a strong impact on the final sample size. Eligibility criteria were: (a) the mother–child dyad to be in good health and (b) mothers to be able to communicate effectively in Greek and have a permanent telephone number for the follow-up process. Initially, 1000 mothers were approached and asked to participate in the study. Of these, 910 agreed to participate (response rate 91%). Finally, 847 mothers were included, as 63 women were excluded due to non-response at one of the three follow-up time points (Figure 1).

### 2.3. Data Collection

Written informed consent was obtained from eligible mothers following a standard, consistent study description. Information was collected through an in-person structured interview conducted by the first author on the third day postpartum. This method ensured completeness and clarity of responses and created familiarity with the researchers which further assisted in maintaining contact with the mothers for the follow-up. Explanations were provided to participants when necessary. Data on medication use during hospitalization were collected from the patient’s medical record.

During the follow-up process, women were contacted by telephone at one, three and six months postpartum to obtain valuable and relevant details regarding their breastfeeding status, reasons for breastfeeding cessation and medication intake throughout this period. Additionally, information on possible complications that led women to use medications was explored and in case of medication intake due to a chronic disease this was also recorded. A code was automatically allocated to each participant by the database used to secure de-identification.

### 2.4. Measurements

The first author designed a structured questionnaire, divided into three sections, for the needs of the study after a thorough review of the relevant literature [2,6,18,19]. The draft instrument was first evaluated by five experts and then pilot-tested on 50 women (not included in the current study sample) to identify potential shortcomings before being implemented in the full study. The final form was approved by the research team and consisted of open-ended and closed-ended questions.

The questionnaire’s first part addressed demographic and socioeconomic characteristics (maternal age, nationality, marital and employment status, educational level, type of hospital, area of residence). The second part provided information on the medical, maternity and lactation history (parity, mode of delivery, gestational age, newborn’s sex and birth weight, breastfeeding duration of a previous child, 6-month follow-up of breastfeeding, breastfeeding duration, reason for breastfeeding cessation, maternal body mass index prior to pregnancy and delivery). The third part included information on maternal medication intake during hospitalization. In addition, records were kept of medications administered due to chronic diseases or complications occurred during the 6-month follow-up. Medications were classified according to the WHO Anatomical Therapeutic Chemical (ATC) classification system. The ATC system is divided into five different levels; in this study, we focus on level 1 (anatomical group), level 2 (therapeutic group) and level 5 (chemical substance) [20].

### 2.5. Drug Classification

Medications used by mothers were classified according to the risk in breastfeeding by the Drugs and Lactation Database—Lactmed [21] and the Hale’s lactation risk categories [22], which are among the most cited risk classifications of medicines in breastfeeding. Lactmed is produced by the National Library of Medicine and Dr Hale is a world-renowned expert in perinatal pharmacology.

Lactmed is a database which includes information on medicines and other chemical substances to which breastfeeding mothers may be exposed [21]. Additional information is provided on the levels of medications in breast milk and infant blood, as well as on the potential adverse effects in the nursing infant. Lactmed does not have standardized classification categories. For this reason, the following descriptive categories have been created:Lactmed 1 includes the categories: “Compatible”; “Limited Data (LD). Compatible”;Lactmed 2 includes the categories: “Probably Compatible”; “Limited Data (LD). Probably Compatible”; “Probably Compatible. May reduce milk supply”;Lactmed 3 includes the categories: “Limited Data (LD). Use alternative drugs”; “No Data (ND). Use alternative drugs”; “Use alternative drugs. May suppress lactation”;Lactmed 4 includes the categories: “No Data (ND). Use with caution. Avoid during lactation”; “Avoid during lactation. Potential toxicity to the infant”.

Hale’s classification [22] includes the following categories:
Level 1 (L1): compatible;Level 2 (L2): probably compatible: Drug that has been studied in a limited number of breastfeeding women, without an increase in adverse effects in the infant. And/or the evidence of a demonstrated risk which is likely to follow the use of this medication in a breastfeeding woman is remote;Level 3 (L3): probably compatible: There are no control studies in breastfeeding women; however, the risk of untoward effects to a breastfed infant is possible, or controlled studies show only minimal non-threatening adverse effects. Drugs should be given only if the potential benefit justifies the potential risk to the infant. (New medications that have absolutely no published data are automatically categorized in this category, regardless of how safe they are);Level 4 (L4): potentially hazardous: There is positive evidence of risk to a breastfed infant or to breast milk production, but the benefits of use in breastfeeding mothers may be acceptable despite the risk to the infant;Level 5 (L5): hazardous.

Taking into account that each source classifies drugs differently in terms of compatibility, in order to be able to compare these sources, we assumed that there is agreement between the categories in the following way:Lactmed 1, Lactmed 2, L1 (Hale) and L2 (Hale);Lactmed 3, L3 (Hale);Lactmed 4, L4 (Hale) and L5 (Hale).

In a similar way, Silveira et al. grouped Hale’s L1 and L2 categories and considered them “compatible”, L3 was considered “judicious” and L4 and L5 “contraindicated” [23]. Based on this assumption, we identified which medicines show a discrepancy in classification in regard to breastfeeding risk. However, we should note that in Lactmed’s classification, the drug is sometimes classified as “probably compatible” and combined with the phrase “use alternative drugs”. In this case, the designation “use alternative drugs” prevails and the categorization of the risk for breastfeeding is based on it. In addition, the L3 category in Hale’s classification includes new medications with no published data. If Lactmed labels the same drug as “No Data. Probably Compatible”, it is assumed that there is no discrepancy in classification between the two sources for the drug.

### 2.6. Data Analysis

Qualitative variables are presented as absolute and relative frequencies (%), while quantitative variables are presented as mean ± SD. Chi-square test was implemented to investigate the relationship between qualitative variables. Mann–Whitney test or Kruskal–Wallis test was implemented to compare two or more subgroups, respectively. To investigate factors that affect breastfeeding cessation due to medication intake, logistic regression analysis was used; towards this end, we investigated the effect of various factors (e.g., marital status, mode of delivery, employment prior to pregnancy, nationality, whether there was a physician’s recommendation to discontinue breastfeeding, duration of previous breastfeeding experience in days, employment at sixth month after delivery, etc.) on breastfeeding cessation due to medication intake. Additionally, multiple regression analysis was applied, where additional and confounding factors were taken into account.

A two-sided test with a *p*-value less than 0.05 was considered statistically significant. IBM SPSS v.28 (IBM Corp. Released 2021. IBM SPSS Statistics for Windows, Version 28.0. Armonk, NY, USA: IBM Corp.) was used for the statistical analysis.

## 3. Results

### 3.1. Basic Sample Characteristics

A detailed description of maternal baseline sociodemographic characteristics and the history related to pregnancy and childbirth can be found in a previous published article [19] and are summarized here as follows:

Of the 847 women who participated in the study, the majority were Greek (91.4%), married (95.4%), with a high level of education (College/University: 64%, Postgraduate studies: 16.8%) and employed before pregnancy (80.9%). The mean age of the studied sample was 33.7 years (33.7 ± 4.7) and almost half of the women were primiparous (52.3%). Approximately half of the women gave birth in a private hospital (54.8%) and a relatively high cesarean section rate was observed (66.8%). The mean duration of pregnancy was 38.3 ± 1.5 weeks. The mean maternal BMI was 24.2 kg/m^2^ (24.2 ± 5.0) before pregnancy and 28.6 kg/m^2^ (28.6 ± 4.8) before delivery. As for the baseline characteristics of the infants (*n* = 859), most of them were full-term (90.3%), had normal birth weight (>2.5 kg the 92.5%; 3134.4 ± 451.2 g), and approximately half of them were male (51.8%).

### 3.2. Maternal Medication Intake during Postpartum Period

#### 3.2.1. Compatibility with Breastfeeding

Overall, 114 different medicines were documented during the women’s hospitalization and six-month postpartum period, 8 of which had some degree of discrepancy in classification with regard to breastfeeding risk between the two sources used, Lactmed and Hale (Appendix A). Appendix A shows the percentages of medications used during the postpartum period and their classification in terms of compatibility with breastfeeding according to Lactmed and Hale. From the 114 medicines, based on Lactmed classification, 35 (30.7%) were considered compatible with breastfeeding, 37 (32.4%) as probably compatible, 18 (15.8%) as judicious and the use of alternative drugs was recommended, 4 (3.5%) to be avoided or to be used with caution and 20 (17.5%) were not classified as the information was not available. According to Hale’s classification, 16 (14%) were included in the safest category (L1); 46 (40.3%) in the safe category (L2); 32 (28%) were classified as moderately safe (L3); 4 (3.5%) as potentially hazardous (L4); 1 (0.9%) as hazardous (L5) and 15 (13.2%) were not classified as information was not available. All medicines used by lactating mothers (796 mothers out of 847) are extensively demonstrated in Appendix A.

#### 3.2.2. Medication Intake as a Reason for Breastfeeding Cessation

Medication intake, after the perceived insufficient milk supply (45.3%, *n* = 180/397), was the second most frequent reason for discontinuation of breastfeeding among the studied population (14.4%, *n* = 57/397). All reasons for breastfeeding cessation have been extensively demonstrated in a previous study [17]. Of the 57 cases of breastfeeding cessation due to medication intake, for 42 women, the decision to cease breastfeeding was based on the counseling of a physician (either an obstetrician or the medical specialist responsible for the treatment of the chronic disease or acute incident) and for 9 mothers was based on their fears of harming their infant. For 6 mothers, there was a reduction in milk supply after a physician’s recommendation to discard the breast milk or reduce the frequency of breastfeeding (Appendix A).

Among mothers who did not initiate or discontinued breastfeeding on their own decision (*n* = 9), 3 of them used escitalopram and ceased breastfeeding despite the recommendations of HPs to continue; 2 mothers reported high doses of methylprednisolone and the first one decided not to initiate breastfeeding after being counseled by her physician that high doses are incompatible with breastfeeding. The second mother discontinued breastfeeding based on internet information; 1 mother was taking methyldopa and nifedipine and decided to discontinue breastfeeding contrary to HPs information on the compatibility of the medication with breastfeeding; 1 woman was using clindamycin, doxycycline and piperacillin-tazobactam and it was her own decision to cease breastfeeding. In the case of clindamycin, Lactmed describes it as probably compatible but suggests the use of alternative medicines, whereas according to Hale’s classification it is probably compatible (L2). As far as methylprednisolone is concerned, in the two aforementioned cases 8–12 mg per day was used. According to Hale, the amount of steroids transferred into human milk is minimal as long as the dose does not exceed 80 mg daily [22]. Moreover, Lactmed reports that oral doses of methylprednisolone less than 1 g do not require special precautions [21]. Finally, 2 women received methadone and buprenorphine-naloxone due to previous illicit drug abuse and decided not to initiate breastfeeding. Fear of harming their infant/newborn led the 88.9% (*n* = 8/9) of mothers to cease breastfeeding by choice, despite the fact that the medication was compatible with breastfeeding. The only case of medication use identified as judicious was that of a mother who used clindamycin and decided to stop breastfeeding, even though she could have taken measures to minimize the infant’s exposure.

Regarding mothers who were advised by a physician to discard their milk due to medication intake (*n* = 6), 3 women were receiving medicines compatible with breastfeeding (gentamicin-dexamethasone topical, methyldopa, amoxicillin and clavulanic acid, diclofenac) and 3 women received medications identified as judicious (metronidazole, ciprofloxacin, clindamycin). Five out of 6 mothers followed the physician’s advice but due to reduction in milk supply they ended up ceasing breastfeeding, while the other one decided to discontinue from the beginning. Erroneous physician’s advice led 3 out of 6 mothers to take measures to minimize infant’s exposure that were unnecessary as the medicines were compatible.

As for women who discontinued breastfeeding due to medication intake following a physician’s recommendation (*n* = 42), 5 women used medicines classified as possibly hazardous or hazardous, 21 women received medicines identified as judicious or not classified or to be used with caution and 15 women were treated with medicines compatible with breastfeeding. One woman who had a car accident was hospitalized and received medications she cannot recall, including antiepileptics for which HPs advised her to discontinue breastfeeding. It is worth noting that in the case where a woman took paroxetine, although classified as compatible, the neonate developed neutropenia and the recommendation to discontinue breastfeeding was after this finding. It is also important to mention that there is a discrepancy in the categorization of valproic acid between the two sources, with Hale describing it as possibly hazardous (L4) due to the risk of neurobehavioral complications to the child, and Lactmed as possibly compatible with the indication to monitor the infant for side effects. Inappropriate physician’s consultation led 50% (*n* = 21/42) of the women to discontinue breastfeeding, although measures to minimize the exposure to judicious medicines could have been taken. Finally, 93.3% (*n* = 14/15) of the women using compatible medicines were erroneously advised to cease breastfeeding.

We conclude that from the 57 cases of breastfeeding cessation due to medication intake, only 10.5% of the cases (*n* = 6/57) received evidence-based counseling from a physician for breastfeeding cessation. Regrettably, 12.3% (*n* = 7/57) of mothers discontinued breastfeeding by choice or due to breast milk reduction despite the fact that they received appropriate professional counseling to maintain lactation. On the contrary, the results of the study deduce that 68.4% (*n* = 39/57) of the participants ceased breastfeeding as a consequence of erroneous professional advice. The remaining 7% (*n* = 4/57) was their personal decision to discontinue breastfeeding without having received any professional advice and 1.8% (*n* = 1/57) involved ambiguous recommendations because inadequate information was given about which medicines were administered.

#### 3.2.3. Factors Associated with Breastfeeding Cessation due to Medication Intake

A remarkable result that emerged from our study was the analysis of the factors associated with breastfeeding cessation due to medication intake. Employing univariate logistic regression analysis, we found factors significantly correlated with breastfeeding discontinuation due to medication intake (Table 1). In detail, the better the mother’s educational level, the less likely she was to discontinue breastfeeding due to medication intake (*p* = 0.034). In terms of non-employment at 6 months postpartum due to the COVID-19 pandemic, fewer women compared to the expected ones, who were on work suspension or teleworking, discontinued breastfeeding due to the use of pharmaceuticals (*p* < 0.001). Additionally, women who had given birth via cesarean section had an OR = 2.8 (95% CI: 1.4–5.8) to discontinue lactation due to pharmaceuticals compared to women who had a vaginal delivery (*p* = 0.005). As far as previous breastfeeding experience is concerned, this was significantly lower for women who discontinued breastfeeding due to medication intake (median= 40 days) compared to women who did not wean for the same reason (median= 181 days) (*p* < 0.001). In addition, mothers who were taking medicines for chronic diseases had an OR = 4.6 (95% CI: 2.6–8.2) to cease breastfeeding compared to those who were not on medication for a chronic disease (*p* < 0.001). Lastly, the physicians’ recommendations to discontinue breastfeeding due to medication intake was a factor that significantly increased the rate of breastfeeding cessation compared to women who did not have such professional advice. No significant differences were observed between breastfeeding cessation due to medication intake and other variables assessed (maternal age, nationality, parity, employment before pregnancy).

Applying multiple logistic regression to the aforementioned variables and adding socioeconomic, lifestyle and behavioral factors that could act as confounders (marital status, type of hospital (private or public), duration of pregnancy, smoking before pregnancy, fear of harming the infant as a reason for refusing to take medications during lactation, personal attitude towards medication intake), the results of the study revealed that among the factors that univariate analysis showed to be significant, physician’s recommendation to discontinue breastfeeding due to medication intake and women who had given birth via cesarean section continued to be significant by applying multiple regression analysis. All factors were examined for possible interactions, but no significant results were detected (Table 2).

## 4. Discussion

The present study, to our knowledge, is the first Greek study that reveals data on the safety of the medicinal products used by puerperal women and reports as well, the effect of professional counseling on maternal decision to initiate or discontinue breastfeeding when the need for medication intake arises.

With respect to the sources used to classify medicines according to the risk during lactation, Lactmed and Hale are both evidenced-based, fully referenced, frequently updated and have been used in pertinent studies [4,7,23,24,25,26]. The present study showed that, according to the Lactmed and Hale classification, respectively, 63.1% and 54.3% of the medicines were considered compatible or probably compatible, 15.8% and 28% were characterized as judicious, 3.5% and 4.4% referred to medications to be avoided and finally 17.5% and 13.2% were not classified as no information was available. Administration of medicines identified as potentially hazardous during lactation was followed by breastfeeding cessation in our study, except from the case of meperidine, classified as potentially hazardous (L4) according to Hale and judicious according to Lactmed, which was administered as a postoperative analgesic to 0.7% (*n* = 6) of lactating women. In a recent study conducted in Brazil based on Hale risk categorization, similar rates to the aforementioned categories were found (50%, 24.4%, 5.8%, 19.6%, respectively) [24]. Compared with older studies that used various databases for risk categorization, we observe that the percentage of medication usage without available data decreases over time, indicating an ongoing attempt to update information on medication intake during lactation (35.8–37.2%) [25,26]. Very recently, Fomina et al. evaluated the strength of evidence for the most usual medications administered to lactating women. Based on the Lactmed database, they showed that only 10% of the medicines had no available data from research studies [27]. It is suggested that, in the future, prominence may be given to research methods that do not involve living patients (in vitro studies or non-human subject research) and to pharmacokinetic models in the hope that the dearth of knowledge will be filled in [27].

Medication intake during lactation is a common reason for non-initiation [28] or lower duration of breastfeeding, strongly associated with inappropriate professional counseling or maternal concerns that medicines may be harmful to their child [5,6,29]. In the present study, 35% (*n* = 20/57) of the participants decided not to initiate breastfeeding due to medication intake which was also the second most frequent reason for the discontinuation of breastfeeding among the studied population (14.4%, *n* = 57/397), findings that are in line with previous studies. For instance, in a Brazilian study, 10.2% of the women reported that they did not initiate breastfeeding because of some medicines they needed to use, while 2.8% of the participants ceased breastfeeding for the same reason [24]. Additionally, de Waard et al. [28] reported that 38.2% of the participants decided not to initiate breastfeeding due to medication intake and in the study by Chaves et al. [25] the use of medication was the fourth most common reason (4.5%) for the discontinuation of breastfeeding. It should be considered though that studies including medications differ significantly, especially in study design (e.g., classification, medicines included, longitudinal or retrospective design, etc.), which can partially explain any prevalence disparities.

It is worth mentioning that among the factors associated with breastfeeding cessation due to medication intake were educational level, employment at six months and medication intake for a chronic disease. Higher educational level was associated with a lower likelihood of breastfeeding discontinuation, as has been shown earlier by other research teams [24]. Non-employment at six months postpartum had a positive effect on the duration of breastfeeding as it decreased the possibility for cessation due to medication intake. Interestingly enough, the results of our study demonstrated that women on work suspension or teleworking, due to pandemic restrictions, were less likely to cease breastfeeding due to medication intake. Published studies show that mothers felt breastfeeding was protected during lockdown due to delayed return at work, while for others the COVID-19 era created more anxiety because of restricted medical and lactation support in-person [17,30,31]. Overall, the impact of the pandemic on breastfeeding is complex and can vary depending on several factors. It is of utmost importance that HPs and policy makers continue to support breastfeeding during and after the tremendous period of a pandemic and ensure that mothers have access to the resources and support they need to breastfeed successfully, including tele-visits. Lastly, mothers who were taking medicines for chronic diseases were more likely to cease breastfeeding compared to those who were not on medication for the same reason. Lactating women with a pre-existing chronic condition have a higher likelihood of medication use during lactation [16]. A very recent study conducted in Canada concluded that breastfeeding duration was significantly shorter for chronically ill women who discontinued their medications while breastfeeding, and their breastfeeding goals were less likely to be accomplished compared to women who continued to take their medicines [32]. Thus, future studies should particularly examine the impact of medication use due to chronic diseases on breastfeeding initiation, duration and cessation.

When multiple logistic regression analysis was applied, smoking before pregnancy was found to be a significant factor for breastfeeding cessation due to medication intake. Previous studies have highlighted the association between smoking before pregnancy and early discontinuation of breastfeeding [33,34]. The fact that some factors were found to be significant in univariate analysis but not significant in multiple logistic regression may be due to the small number (*n* = 57/847) of the study population that ceased breastfeeding due to medication intake and does not underestimate the significance of factors when analyzed in a univariate model.

Another issue of concern revealed in this study was that a large proportion of mothers (68.4%, *n* = 39/57) were led to weaning after inappropriate counseling by a physician, a finding consistent with previous results. In a study where women required propylthiouracil postpartum, 33% of the physicians (endocrinologists, family physicians, pediatricians) erroneously advised women not to breastfeed while 5% of them gave equivocal advice [35]. Another study including mothers with systemic lupus erythematosus in Argentina found that 41% of those who ceased breastfeeding early did so because of medication usage [36]. Participants reported reduced rates of breastfeeding initiation and duration and the reason for discontinuation was often attributed to therapies that were actually of low risk. The inconsistency of different physicians may increase the likelihood of breastfeeding cessation and therefore impact negatively on mothers’ breastfeeding goals. Despite the presence of nationwide helplines, the Greek medical personnel provided considerable misinformation about medicines and breastfeeding [11,12]. In a personal communication with a midwife in charge of the “Alkyoni” helpline named CE, she reported that according to their annual statistics, compatibility of medications with breastfeeding is among the top three issues of telephone calls from lactating mothers. However, HPs rarely call to obtain information on medications although they are the ones who usually recommend the helpline to mothers. Nevertheless, in our study, none of the women who discontinued breastfeeding due to medications used the established helplines to seek information. Only 0.4% of the total study sample reported that they would refer to helpline support if the need for medication intake arose during lactation. Therefore, promotional campaigns should be carried out in this area.

Measures that could be taken to minimize infant exposure to medication through breast milk include the administration of an alternative medicine at the lowest effective dose, with a short half-life, high protein binding, low oral bioavailability or high molecular weight [21,22,37]. Other measures include administering the maternal dose after breastfeeding and before the infant’s longest sleep, so that the medicine’s concentration has decreased by the time of the next feeding, except in the case of drugs with a long half-life; avoidance of breastfeeding or discarding breast milk for a certain period of time (e.g., 4 h) after the maternal dose, which will significantly reduce the infant’s exposure [21,22]. If the mother should follow a long-term drug treatment, she might consider setting up a milk bank in advance so that she can feed the infant safely while taking the medication. Previous evidence has shown that women taking most of the study-mentioned medicines, breastfed their infants and low or undetectable infant drug plasma levels were reported, minor or no adverse effects were noted in the infants and normal weight gain was observed. Nevertheless, it is recommended that infants should always be monitored for side effects, each mother is individualized according to the medicine and her condition and shared decision-making should be supported.

Globally, various online resources and printed drug monographs provide HPs with information about safety and compatibility of medicines during lactation [21,22]. Therefore, HPs do have access to recent scientific contents so as to provide evidence-based information to mothers and aid the shared decision-making process regarding breastfeeding initiation, maintenance and cessation. In the absence of clear and sufficient evidence, experts in perinatal pharmacology should be sought. Although in our study only one infant experienced adverse side effects (neutropenia), we propose that emphasis should be given on the population pharmacokinetic (pop PK) modeling which provides evaluations on infant drug exposure via breast milk through simulation [7]. Furthermore, disappointingly in our study, mothers did not include midwives among the HPs who were asked for advice on the compatibility of medicines with breastfeeding. It appears that the Greek population is not familiar with the fact that the midwife’s role in supporting lactation also entails medication counseling. Lastly, we strongly believe that midwives can critically contribute to pharmacovigilance systems and detect safety signals from the use of medicines during the perinatal and neonatal period. The detection of adverse drug reactions and their evaluation through pharmacovigilance procedures will definitely provide essential information and update healthcare databases [38].

This study has gone some way towards enhancing our understanding of medication intake as a factor for non-initiation or breastfeeding cessation during the COVID-19 pandemic, especially in the Greek setting. We are aware, though, that our research had some strengths and limitations. A strength of the present study is the use of two classifications—Lactmed and Hale’s—which are among the most cited risk classifications of medications during lactation. Furthermore, by categorizing the medicinal products according to the ATC classification system, which is a uniform coding rule, the validity of the findings was increased, and it was possible to compare them with other studies having used the same classification system. Another strong point of our work lies in the fact that information concerning medication use during hospitalization was collected from the patient’s medical record; this record provided data more impartially compared to the interview used during the follow-up process, which could be subject to an inherent recall bias and lack of accuracy. However, we endeavored to reduce recall bias by using open-ended questions which provide extra details and actionable insight for the researchers. An additional limitation to be considered is that our analysis does not include the frequency of use or dosage of all the medicines used. Finally, the high prevalence of participants’ educational level (80.8% tertiary education) might have influenced their attitudes towards medication intake. Higher education has been associated with a higher rate of medication use during lactation in other studies [2,24]. Therefore, our study may have overestimated the actual rates of medication use. Lastly, although the study sample derived from five different hospitals both public and private, they were all located in a single prefecture and therefore the results cannot be generalized to the whole Greek context.

## 5. Conclusions

Medication intake is demonstrated as a significant reason for non-initiation and cessation of breastfeeding. Multiple factors, including inconsistent and improper professional counseling regarding medication intake, are importantly associated with increased breastfeeding cessation. Furthermore, uncertainty about the extent to which SARS-CoV-2 could be transmitted through human milk in the first half of 2020 might have contributed to increased breastfeeding discontinuation rates. Interestingly, the COVID-19 pandemic appeared to have reduced breastfeeding cessation due to medication intake in Greece probably due to lockdowns, home confinement and teleworking. We conclude that maternal, infant and lactation consultancy in Greece should be based on regularly reviewed guidelines, contemporary databases and be strictly related to evidence-based approaches. Implementation of initiatives, consulting services and continuing educational programs oriented towards HPs and mothers with reflection on the topic of medication intake during lactation could positively affect breastfeeding indicators in Greece.

## Figures and Tables

**Figure 1 children-10-00586-f001:**
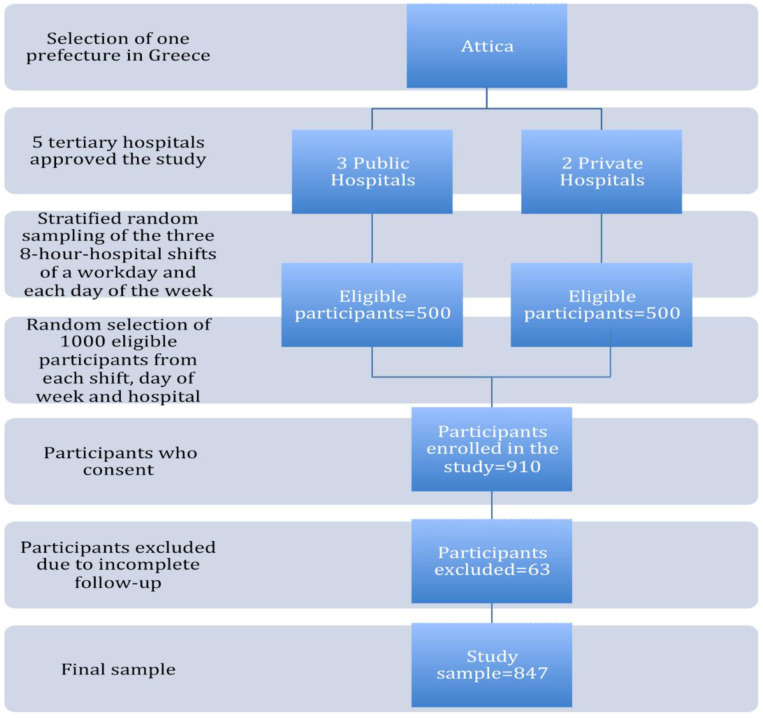
Flowchart of the multistage stratified random sampling method used in the study. (Attica/Greece 2020; *n* = 847).

**Table 1 children-10-00586-t001:** Univariate logistic regression analysis for breastfeeding cessation due to medication intake. (Attica/Greece 2020; *n* = 57/847).

BF Cessation due to Medication Intake	OR (95% CI)	*p*-Value
Qualitative variables		
Physician’s recommendation for BF cessation due to medication intake On physician’s recommendation Women without recommendation by a physician to discontinue BF	93.9 (45.6–193.4) Ref	<0.001
Medication intake for chronic disease Women who received medication for chronic diseases Women who did not receive medication for chronic diseases	4.6 (2.6–8.2) Ref	<0.001
Mode of delivery Cesarean section Vaginal delivery	2.8 (1.4–5.8) Ref	0.005
Non-employment at 6 months after delivery Not employed Maternity leave Work suspension due to COVID-19 pandemic Teleworking due to COVID-19 pandemic Mothers working at 6 months postpartum	0.54 (0.27–1.06) 0.10 (0.02–0.40) 0.08 (0.01–0.62) 0.17 (0.05–0.54) Ref	<0.001
Educational level College University Postgraduate studies Women with education up to High School	0.7 (0.3–1.6) 0.6 (0.3–1.1) 0.2 (0.1–0.6) Ref	0.034
Nationality Greek Others	Ref	0.467
Employment before pregnancy Women working before pregnancy Women not working before pregnancy	Ref	0.727
Quantitative variables		
Previous BF experience (days)	0.994 (0.992–0.997)	<0.001
Maternal age		0.444
Parity		0.601

BF: breastfeeding; OR: odds ratio; CI: confidence interval.

**Table 2 children-10-00586-t002:** Multiple logistic regression analysis for breastfeeding cessation due to medication intake. (Attica/Greece 2020; *n* = 57/847).

BF Cessation due to Medication Intake	OR (95% CI)	*p*-Value
Physician’s recommendation for BF cessation due to medication intake On physician’s recommendation Women without recommendation by a physician to discontinue BF	171.44 (67.401–436.075) Ref	<0.001
Mode of delivery Cesarean section Vaginal delivery	6.097 (2.062–18.025) Ref	0.001
Type of hospital Private Public	2.261 (0.981–5.209) Ref	0.055
Smoking before pregnancy (cigarettes/day)	1.103 (1.059–1.149)	<0.001

BF: breastfeeding; OR: odds ratio; CI: confidence interval.

## Data Availability

No new data were created or analyzed in this study. Data sharing is not applicable to this article.

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
