# Peer review of "Medication Intake as a Factor for Non-Initiation and Cessation of Breastfeeding: A Prospective Cohort Study in Greece during the COVID-19 Pandemic"

_children, 2023, doi:10.3390/children10030586_

Round 1

Reviewer 1 Report

In the paper Medication intake as a factor for non-initiation and cessation of 2 breastfeeding. A prospective cohort study in Greece during the 3 COVID -19 pandemic, the authors investigate maternal medication intake as a factor for non-initiation and cessation of breastfeeding and the effect of professional counseling on mother’s decision making. Overall, the review is well written and could be of use for clinicians of different specialties involved in lactation consultancy.

In table 3 the authors summarize cases of breastfeeding cessation due to medication intake and professional counseling.

Cases no 2, 5, 6, 8, 9, 12, 15, 26, 30, 32, 33, 34, 50, 51, 54: In all these cases the mothers received medication that was identified as judicious. The authors argue that breastfeeding should have continued, with measures to minimize the exposure of the infants, however, they do not specify what these measures are. It seems appropriate to specify for each case, what those measures should be.

Case no 17: Oxcarbazepine labeled according to Lactmed “Probably compatible. Use with caution” and L3 by Hale. However, the authors argue that both the neurologist and obstetrician gave “erroneous counseling due to compatibility of the medicine with BF”.  The authors should detail why “Use with caution” is considered compatible!

Case no 35: Ambroxol is not classified in either of the 2 databases, yet the authors seem to consider this to be equivalent to safe for breastfeeding. The authors should clarify this.

Author Response

Journal

 Children

Manuscript ID

children-2240182

Manuscript Title 

Medication intake as a factor for non-initiation and cessation of breastfeeding. A prospective cohort study in Greece during the COVID-19 pandemic.

The authors would like to express their gratitude to the reviewer for his/her useful comments that helped to improve further our manuscript. Please find below the authors’ responses to the reviewer’s comments.

Reviewer 1

Comment

Response

In table 3 the authors summarize cases of breastfeeding cessation due to medication intake and professional counseling.

Cases no 2, 5, 6, 8, 9, 12, 15, 26, 30, 32, 33, 34, 50, 51, 54: In all these cases the mothers received medication that was identified as judicious. The authors argue that breastfeeding should have continued, with measures to minimize the exposure of the infants, however, they do not specify what these measures are. It seems appropriate to specify for each case, what those measures should be.

We thank the reviewer for his/her comment!

We have now added a paragraph describing, for each case presented in Table 1 (Table’s numbering has changed based on comments by reviewer 2), measures to minimize infant exposure to medicines through breast milk. (Please see Subsection 3.2.2, 2nd paragraph).

Case no 17: Oxcarbazepine labeled according to Lactmed “Probably compatible. Use with caution” and L3 by Hale. However, the authors argue that both the neurologist and obstetrician gave “erroneous counseling due to compatibility of the medicine with BF”.  The authors should detail why “Use with caution” is considered compatible!

We thank the reviewer for this point. 

Instead of “Erroneous counseling due to compatibility of the medicine with BF”, we should have written “Erroneous counseling. The HP could have advised the mother to take measures to minimize the infant’s exposure”.

We have now altered the sentence (see Table 1, Case 17) and have corrected the numbers/percentages in the main text following this change (Subsection 3.2.2: 1st and 5th paragraph).

Case no 35: Ambroxol is not classified in either of the 2 databases, yet the authors seem to consider this to be equivalent to safe for breastfeeding. The authors should clarify this.

We thank the reviewer for his/her thorough review!

Note: Ambroxol is a mucolytic of choice, widely used and well tolerated during breastfeeding [26]. It is also used in infants [27] and children [28].

We have now added the clarification to Table 1, Case 35.

Reviewer 2 Report

The study investigated maternal medication intake as a factor for non-initiation or cessation of breastfeeding. Mothers were contacted in the newborn nursery and at ages 3 and 6 months.      Human milk is considered the best form of early-life nutrition with short- and long-term benefits for health and development. It is therefore essential to monitor the safety of breastfeeding. The study period was 1-12. 2020. The researchers mention the COVID-19 pandemic briefly in the abstract and provide short comments in the introduction and discussion sections. However, until August 19, 2020, it was not clear if the active replication-competent virus could be transmitted through human milk which influenced more than any other factor (e.g. medications) initiation and continuation of breastfeeding. Half a year of uncertainty resulted in fear-based confusion, misinformation, and increased risk of breastfeeding cessation. It is questionable if phone interviews in intervals of 3 months can precisely evaluate the quality and details of information in such a turbulent time period. The reviewer is sure that the researchers have some data on COVID-19-related behavior changes in the study cohort. This information should be added to the confounders.   Statistical analysis: the study was done in a highly selective cohort and results cannot be generalized. Confounders (e.g. socioeconomic and lifestyle factors) should be further analyzed employing multiple regression analysis because there might be significant interactions.     The study indicated significant misinformation on medications and breastfeeding provided by medical personnel in Greece. Most EU - countries have established centralized phone-based qualified medical information on the potential toxicity of medications where medical personnel and mothers can get information around the clock. The situation in Greece should be described in the introduction section.      Tables 1 and 2 are long and very detailed - should go to the supplementary materials.

Author Response

Journal

 Children

Manuscript ID

children-2240182

Manuscript Title 

Medication intake as a factor for non-initiation and cessation of breastfeeding. A prospective cohort study in Greece during the COVID-19 pandemic.

The authors would like to express their gratitude to the reviewer for his/her useful comments that helped to improve further our manuscript. Please find below the authors’ responses to the reviewer’s comments.

Reviewer 2

Comment

Response

The researchers mention the COVID-19 pandemic briefly in the abstract and provide short comments in the introduction and discussion sections. However, until August 19, 2020, it was not clear if the active replication-competent virus could be transmitted through human milk which influenced more than any other factor (e.g. medications) initiation and continuation of breastfeeding. Half a year of uncertainty resulted in fear-based confusion, misinformation, and increased risk of breastfeeding cessation.

We thank the reviewer for his/her comment!

We have now added extra paragraphs in which we present more extensively the COVID-19 pandemic. Please see the Introduction (5th paragraph), the Discussion (4th paragraph) and the Conclusions.

It is questionable if phone interviews in intervals of 3 months can precisely evaluate the quality and details of information in such a turbulent time period.

Authors followed up women at the 1st, 3rd and 6th month postpartum.  Consequently, the intervals of 3 months were in fact much less. COVID-19 restrictions prevented authors from home visits so phone interviews were a mandatory research tool. Moreover, similar studies have based their follow up on phone interviews [Chaves et al., 2011; de Waard et al., 2019; Stultz et al., 2007] and the high response rate in our study strengthens its results.  

Chaves, R.M.; Lamounier, J.A.; César, G.C. Association between duration of breastfeeding and drug therapy. Asian Pac J Trop Dis. 2011,1, 216-221. doi.org/10.1016/S2222-1808(11)60032-7

de Waard, M.; Blomjous, B.S.; Hol, M.L.F.; Sie, S.D.; Corpeleijn, W.E.; van Goudoever, J.H.B.; van Weissenbruch, M.M. Medication Use During Pregnancy and Lactation in a Dutch Population. J Hum Lact. 2019, 35, 154-164. doi:10.1177/0890334418775630

Stultz, E.E.; Stokes, J.L.; Shaffer, M.L.; Paul, I.M.; Berlin, C.M. Extent of medication use in breastfeeding women. Breastfeed Med. 2007, 2, 145-151. doi:10.1089/bfm.2007.0010

The reviewer is sure that the researchers have some data on COVID-19-related behavior changes in the study cohort. This information should be added to the confounders. Statistical analysis: the study was done in a highly selective cohort and results cannot be generalized. Confounders (e.g. socioeconomic and lifestyle factors) should be further analyzed employing multiple regression analysis because there might be significant interactions.    

We thank the reviewer for his/her thorough review!

-Unfortunately, we did not collect data on COVID-19-related behavior changes (ex. depression scores, anxiety symptoms, illicit substance use etc) as the questionnaire was initially designed in 2019, a period preceding the COVID-19 pandemic. The questionnaire was piloted and IRB approvals were mainly obtained within 2019 (4 out of 5 hospitals) so authors could not proceed with further insertions.  However, other behaviors, such as maternal fear of harming the infant as a reason for refusing to take medications during lactation and personal attitude towards medication intake, were investigated, which could act as confounders. The authors employed multiple regression analysis and these variables showed no interaction. 

-Indeed, we agree that results cannot be generalized and this has been now mentioned in the limitations of the study. Nevertheless, we believe that the results of the study are innovative and interesting and may provide valuable information for future research.

- Authors thank the reviewer for his/her valuable suggestion regarding multiple regression analysis. We have performed multiple logistic regression analysis, employing the suggested confounders and investigating possible interactions. Due to the rather small number of women that ceased breastfeeding due to medication intake (N=57/847), we did not detect many statistical significant variables in the aforementioned analysis. Additionally, all interaction terms were not significant. 

The results from the final model are presented in an additional Table (see Table 3) and are analyzed in the Results (Subsection 3.2.3: 2nd paragraph) and the Discussion section (5th paragraph).  Finally, an addition was made to the abstract and the Data analysis (see Subsection: 2.6: 1st paragraph).

Finally, we believe that the Reviewer’s suggestion was very accurate and has helped us to improve the quality of our submitted manuscript.

The study indicated significant misinformation on medications and breastfeeding provided by medical personnel in Greece. Most EU - countries have established centralized phone-based qualified medical information on the potential toxicity of medications where medical personnel and mothers can get information around the clock. The situation in Greece should be described in the introduction section.

We have now added a new paragraph (see 4th paragraph) in the Introduction section describing the situation in Greece. Additionally, we have commented in the Discussion section on the possible reasons for misinformation about medications and breastfeeding provided by medical personnel in Greece (See 6th paragraph).

Tables 1 and 2 are long and very detailed - should go to the supplementary materials.

Following the reviewer’s comment, authors have  now moved Table 1 and Table 2 to Supplementary materials after being labeled as Table S1 and Table S2 (see Subsection 3.2.1: 1st paragraph).

Round 2

Reviewer 2 Report

The authors responded to all comments of the reviewer and introduced changes in the manuscript. However, the manuscript substantially increased in length (30 pages). Parts of the manuscript are now difficult to read or present details, which are not relevant.

The study's primary outcome was the quality of medical consultations in Greece if mothers had to take drugs during lactation. The authors now indicate that in Greece there is a nationwide phone consultation program (since 2013/2014) where mothers can seek advice if they have to take medications. There is no mention that this medical phone opportunity was used by lactating mothers or their consulted medical professionals. Governments, medical associations, and other NGOs introduce and maintain public health programs. It is important to review those programs in well-designed studies. It is not clear to the reviewer if the study authors asked detailed questions about the quality of the medical resource who provided the advice to continue or stop breastfeeding.

30: type of hospital is not significant in multiple regression analysis: p > 0.05

32: ....The COVID-19 restrictions protected women from ceasing breastfeeding due to medication  intake..... The authors don't show data to support this statement 

299 The new Table 1 describes 57 cases of breastfeeding cessation and comprises 8 (!) manuscript pages. Those details are of minor interest. 

304-334 The newly introduced paragraph includes an interpretation of the data and suggestions on what should be done. This should be shortened and moved to the discussion section. 

Author Response

  Journal

 Children

Manuscript ID

children-2240182

Manuscript Title

Medication intake as a factor for non-initiation and cessation of breastfeeding. A prospective cohort study in Greece during the COVID-19 pandemic.

The authors would like to express their gratitude to the reviewer for his/her useful comments that helped to improve further our manuscript. Please find below the authors’ responses to the reviewer’s comments.

Round 2

Reviewer 2

Comment

Response

"Introduction must be improved"

Reviewer 2 has marked that Introduction "must be improved". It should be mentioned though that authors do have addressed all the requests from Reviewer 2 in the Introduction.

The authors responded to all comments of the reviewer and introduced changes in the manuscript. However, the manuscript substantially increased in length (30 pages). Parts of the manuscript are now difficult to read or present details, which are not relevant.

We agree that the manuscript is rather long, so we reduced its length by transferring 2 tables in the supplementary material as suggested in your previous review. 

Now we have also moved Table 1 in the Supplementary material, following your suggestion, reducing in this way the total length of the manuscript to 16 pages.  

The authors now indicate that in Greece there is a nationwide phone consultation program (since 2013/2014) where mothers can seek advice if they have to take medications. There is no mention that this medical phone opportunity was used by lactating mothers or their consulted medical professionals.

The study included data on which source of information mothers use when it comes to taking medication during breastfeeding. We have now added information about the medical phone opportunity for lactating mothers to the Discussion Section. Please see paragraph 6: “Nevertheless, in our study none of the women… during lactation.”

Unfortunately, we have no data on the use of the medical phone opportunity by the medical professionals who consulted women on breastfeeding cessation due to medication intake as this was out of the scope of the study.

Governments, medical associations, and other NGOs introduce and maintain public health programs. It is important to review those programs in well-designed studies.

It is not clear to the reviewer if the study authors asked detailed questions about the quality of the medical resource who provided the advice to continue or stop breastfeeding.

Unfortunately, there are no Greek published studies regarding these programs. However, authors contacted personally a midwife in charge of the “Alkyoni” helpline, named Chryssa Ekizoglou. She reported useful information that has now been added to the Discussion Section. Please see paragraph 6: “In a personal communication … recommend the helpline to mothers.” 

Τhe quality of the medical resource that provided the advice to continue or stop breastfeeding was out of the scope of this study and was not evaluated. The high proportion (68.4%) of participants who discontinued breastfeeding due to medication intake as a consequence of erroneous professional advice, along with the fact that established support programs are not familiar to lactating mothers and health professionals, highlights the need for further research in a well-designed study.

30: type of hospital is not significant in multiple regression analysis: p > 0.05

We deleted this finding from the abstract and the main text except from Table 2.

32: ....The COVID-19 restrictions protected women from ceasing breastfeeding due to medication intake..... The authors don't show data to support this statement.

In the Results Subsection 3.2.3 (1st paragraph), data are shown to support this statement: “In terms of non-employment at 6 months postpartum due to the COVID-19 pandemic…due to the use of pharmaceuticals (p<0.001).”

299 The new Table 1 describes 57 cases of breastfeeding cessation and comprises 8 (!) manuscript pages. Those details are of minor interest. 

Following the reviewer’s comment, authors have now moved Table 1 to Supplementary materials after being labeled as Table S4. 

304-334 The newly introduced paragraph includes an interpretation of the data and suggestions on what should be done. This should be shortened and moved to the discussion section. 

We have now shortened and moved the paragraph to the discussion section. 

However, as this paragraph contained information on measures to be taken on a case-by-case basis, which was requested by the other reviewer, these data have now been added to table S4, respecting his/her suggestion to improve the information provided in the manuscript.
